# Disaster and distress: The double burden. Depression, anxiety, and post-traumatic stress disorder in doctors during the COVID-19 pandemic in Pakistan

Madah Fatima[1]*, Nazish Imran[2], Irum Aamer[3], Somia Iqtadar[4], Bilquis Shabbir[5]

**1** Academic Department of Psychiatry & Behavioral Sciences, Mayo Hospital, King Edward Medical University, Lahore, Pakistan, **2** Child & Family Psychiatry Department, Mayo Hospital, King Edward Medical University, Lahore, Pakistan, **3** Academic Department of Psychiatry & Behavioral Sciences, Mayo Hospital, King Edward Medical University, Lahore, Pakistan, **4** Department of Medicine, Mayo Hospital, King Edward Medical University, Lahore, Pakistan, **5** Department of Medicine, Mayo Hospital, King Edward Medical University, Lahore, Pakistan

* madahfatima@gmail.com

## Abstract

Healthcare providers are at a high risk of occupational stress, psychological distress, and mental health issues due to unique job demands. The unprecedented negative impact of the COVID-19 pandemic on healthcare further amplified the risk. We aimed to find out the prevalence of anxiety, depression, and post-traumatic stress disorder (PTSD) in doctors during the COVID-19 pandemic, and explore various associated factors. We conducted a cross-sectional survey among clinicians across tertiary care hospitals in Lahore, Pakistan through online forms/ paper-based questionnaires, using the Patient Health Questionnaire (PHQ-9), Generalized Anxiety Disorder (GAD-7) Scale, and The Impact of Events Scale-Revised (IES-R) from July-November 2021. Doctors aged 20–60 years working at the public-sector hospitals of Lahore were included through non-probability convenience sampling. The non-parametric Mann-Whitney U and Chi-square test of independence were applied for inferential data analysis in SPSS version 26 at $\alpha = 0.05$. Of 304 participants, the majority were males (54.6%), from medicine and allied departments (76.9%), junior staff (72%), and front-line workers (75%). 9.5% had a history of psychiatric illness. The prevalence of depression was 25%, anxiety 31.9%, PTSD 15.8%, severe depression 13.8%, and severe anxiety 7.2%. The total median (IQR) scores of depression, anxiety, and PTSD were 5(2-9.5), 4(0–7), and 10(1–23), respectively. Females and junior staff had comparatively severe symptoms of anxiety and depression. Psychiatric history was linked to severe depression (p = 0.003) and PTSD (p=<0.001) but not anxiety (p = 0.136). There were no statistically significant differences in the anxiety and PTSD severity across departments, and between front-line/ second-line work. We found high levels of depression, anxiety, and PTSD in our

**Data availability statement:** Data are available at https://github.com/madahfatima/The-Double-Burden-Study.

**Funding:** The author(s) received no specific funding for this work.

**Competing interests:** The authors have declared that no competing interests exist.

physician sample, even during the 4th COVID-19 wave. This has implications for emphasizing the significance of the mental well-being of healthcare providers and identifying effective interventions to prioritize it even after the pandemic.

## Introduction

The SARS-CoV-2 pandemic posed unmatched challenges to the public health infrastructure globally and caused a catastrophic global economic fallout due to the unpreparedness of healthcare systems to respond to a public health disaster of this intensity. The economic downturns, business shutdowns, and widespread unemployment stemming from the disease burden, disaster mitigation, and disease containment measures, such as lock-downs, had a global impact. The COVID-19 pandemic also seriously impacted healthcare services and healthcare providers by causing economic losses to healthcare facilities, disruption of essential services, and an acute shortage of resources [1]. The healthcare providers faced a threat to their health and lives while working on the front-lines. According to the World Health Organization (WHO), approximately 80,000–180,000 health and care workers (HCWs) were estimated to have died due to COVID-19 pneumonia by May 2021, a number probably much lower than the actual statistics [2]. As per reports by Pakistan Medical Association, more than 200 doctors and 30 paramedics had passed away due to COVID-19 infections in Pakistan by May 2021 [3].

Healthcare providers face unique job demands and workplace stressors, which predispose them to an increased risk of mental distress and illness. Crises such as the COVID-19 pandemic intensify risks to healthcare providers' mental health in addition to endangering their physical health and safety. There is a strong link between compassion fatigue and clinician stress which often results from intense patient interactions and excessive work hours [4]. During the COVID-19 pandemic, healthcare providers experienced an increase in clinical burnout, emotional exhaustion, and compassion fatigue in comparison to pre-pandemic levels, largely driven by depression, anxiety, insomnia, and other mental health problems [5]. Evidence shows that the poor mental health of doctors has far-reaching adverse implications for patient safety and quality of care [6].

Amid the evolving COVID-19 pandemic, there was a growing emphasis on mental health research in the global pandemic response underscoring the need for robust data on its psychological impact on vulnerable populations including healthcare professionals [7]. Evidence from around the world, including several systematic reviews and meta-analyses, accentuated a high magnitude of the adverse psychological impact of COVID-19 on healthcare workers. Andhavarapu, S, and others reported that over one-third of healthcare personnel spanning various roles experienced Post-traumatic stress disorder (PTSD) during the pandemic, with a pooled prevalence of 34–35% [8]. Additional systematic reviews also foregrounded high rates of depression, anxiety, and insomnia with pooled prevalence estimates of 23–28% for depression, 22–33% for anxiety, and nearly 38% for insomnia [9,10]. The major risk factors of psychological distress in healthcare workers were female gender, pre-existing

physical and mental health conditions, exposure to and infection with coronavirus SARS-CoV-2, financial hardships, front-line duties, social isolation, and quarantine measures. On the other hand, family support, adequate healthcare resources, and personal protection were protective factors [10].

The purpose of this study was to assess the presence and severity of depression, anxiety, and post-traumatic stress disorder among doctors across tertiary care hospitals in Lahore during the fourth wave of the COVID-19 pandemic as well as the role of various associated factors such as gender, department,working position, level of seniority, and history of psychiatric illness.

## Materials and methods

### Ethics statement

This study received formal ethical approval from the Institutional Ethical Review Board of King Edward Medical University (KEMU) vide approval letter number 538/ARA/KEMU/2021. Participants who responded through paper-based questionnaires signed a written informed consent, and those completing the online forms acknowledged a consent statement included in the form. This study did not include minors.

### Study design and sample

This study is a cross-sectional survey conducted across tertiary care hospitals in Lahore from 14th of July to 15th of November 2021. We calculated the sample size using Cochran's method with a margin of error set at 5% and a proportion of 0.5, which was estimated to be 383. However, on account of unavailability of official and accurate statistics on the total number of doctors working at the Public sector hospitals in Lahore, finite population correction (that would have reduced the total estimated sample size) was not applied. Non-probability convenience sampling was used to collect data. Doctors in the 20- to 60-year age bracket, employed at public sector teaching hospitals in Lahore, were included. Those working at their respective hospitals for less than three months were excluded. Participants could choose to complete either an online form or a paper-based questionnaire to participate in the study. The online forms were distributed through social media, and paper-based questionnaires were distributed in the departments through the support of mainly consultants working there. Our sample comprised participants largely from three public sector hospitals in Lahore.

### Data collection tools

The 9-item Patient Health Questionnaire (PHQ-9) was used for the assessment of depression and its severity levels, the 7-item Generalized Anxiety Disorder (GAD-7) scale for anxiety, and the Impact of Events Scale-Revised (IES-R) for assessment of PTSD. The three scales have been validated in Pakistan [11–13]. The English versions of the scales were used in our study.

PHQ-9 is a self-report questionnaire that assesses depression over two weeks on a 4-point Likert-scale ranging from 0-3 across the response categories of Not at all (0), Several Days (1), More than Half the Days (2), and Nearly Every day (3) [14]. The total scores on PHQ-9 range between 0 and 27, and interpretation is as follows: 1–4 indicates minimal depression, 5–9 mild, 10–14 moderate, 15–19 moderately severe, and 20–27 severe depression [15]. The PHQ-9 scale has been internationally validated and has demonstrated good internal consistency, sensitivity (78%), and specificity (87%) at cutoff scores of ≥10 for the assessment of depression [16].

We assessed anxiety levels among study participants using the 7-item Generalized Anxiety Disorder tool (GAD-7). GAD-7 measures anxiety levels on a 4-point Likert scale ranging from 0 (Not at All) to 3 (Nearly Every Day). The GAD-7 yields a total score ranging from 0 to 21, with higher scores indicating a greater severity of anxiety. Scores are interpreted as follows: 0–4 (Normal), 5–9 (Mild), 10–14 (Moderate), and 15–21 (Severe) levels of Anxiety. The GAD-7 has 89% sensitivity and 82% specificity for identifying moderate to severe Generalized Anxiety Disorder at cutoff scores ≥10 [17].

The Impact of Events Scale-Revised (IES-R) is a brief, self-report questionnaire comprising items or questions related to the intrusion, avoidance, and hyper-arousal in addition to overall subjective stress response of PTSD, measured on a 5-point Likert Scale (0 = Not at all; 1 = A little bit; 2 = Moderately; 3 = Quite a bit; 4 = Extremely). Although there is no consensus over the cutoff scores, higher scores reflect more severe levels of PTSD, with scores higher than 24 considered significant. The IES-R scoring is interpreted as follows: 24–32 (Partial PTSD), 33–38 (Probable PTSD), and 39 and above as PTSD present ("at levels high enough to suppress immune system functioning even 10 years after an impact event)" [18,19].

## Data analysis

All data were analyzed using IBM Statistical Package for Social Sciences, version 26 (IBM Corp., Armonk, New York). Descriptive statistics were used to summarize study participants' demographic characteristics and the prevalence and severity levels of depression, anxiety, and PTSD symptoms, presented as frequencies *(f)* and percentages *(%)*. Since normality testing revealed that the total PHQ-9, GAD-7, and IES-R scores were not normally distributed, they were reported as Median and Interquartile Ranges (IQRs). We applied the non-parametric Mann-Whitney *U* Test for inferential data analysis to compare median scores of anxiety, depression, and PTSD on the respective scales between two independent variable groups on account of the non-normal distribution of data. The chi-square test of independence was applied to explore the association between various factors and the presence and severity of anxiety, depression, and PTSD. It determined independence between categories of background characteristics and severity levels of anxiety, depression, and PTSD. To account for the Type 1 error risk due to multiple testing, the p-values were adjusted through the Bonferroni correction method to control for multiple comparisons. The significance level for inferential statistics was set at $\alpha = 0.05$.

## Results

### Background characteristics

Three hundred and four (304) doctors took part in the study, of which 138 (45.4%) were females and 166 (54.6%) were males. The mean age of the study participants was 32.1 with a standard deviation of 6.4 years, and the majority (207, 68.1%) were married.

A large majority of participants (n = 234), accounting for 76.9% belonged to the medicine and allied specialties, and 70 (23%) were from the surgery and allied departments. Around 72% (n = 219) held junior positions, and only 85 (28%) identified themselves as senior staff members. 228 respondents (75%) were frontline workers directly engaged in the care of suspected and confirmed cases of COVID-19 pneumonia. One hundred and fifty-six (51.3%) participants had professional experience of less than 5 years. Only a small proportion of the study participants (n = 29, 9.5%) reported a previous history of mental health issues.

### Prevalence of depression, anxiety, and PTSD

The prevalence of depression was 25%, anxiety 31.9%, and PTSD 15.8% (Table 1). The total median (IQR) scores of depression, anxiety, and PTSD were 5(2-9.5), 4(0–7), and 10(1–23), respectively. For PTSD subscale categories, the Median (IQR) scores were 4(0–8) for intrusion, 4(0–10) for avoidance, and 2(0–6) for hyper-vigilance (Table 2). The prevalence of severe depression and severe anxiety was 13.8% and 7.2% respectively (Table 3).

The reliability analysis showed that the Cronbach's alpha values for PHQ9, GAD7, and Impact of Events Scale-Revised (IES-R) were 0.91, 0.92, and 0.97 respectively in our study sample. As shown in Table 2, statistically significant differences were present in the total median scores of depression, anxiety, and PTSD across gender, medicine or surgery departments, different levels of seniority, frontline or second-line working position, and positive or negative history of

**Table 1. Overview of participants' demographic information.**

| Background Characteristics | | |
|---|---|---|
| **Age** | *Mean* | *SD* |
| | 32.1 | 6.4 |
| | *f* | *%* |
| **Gender** | | |
| Male | 166 | 54.6% |
| Female | 138 | 45.4% |
| **Staff's Position/ Seniority** | | |
| Juniors(House officer, Medical officer, Postgraduate Resident) | 219 | 72% |
| Seniors(Senior Registrar, Assistant Prof., Assoc. Prof., Professor) | 85 | 28% |
| **Line of Work during the Pandemic** | | |
| Frontline Work | 228 | 75% |
| Second line Work | 76 | 25% |
| **Department and Specialty** | | |
| Medicine & Allied | 234 | 76.9% |
| Surgical & Allied | 70 | 23.1% |
| **Total Professional Experience** | | |
| Below 5 years | 156 | 51.3% |
| 5 to10 years | 78 | 25.7% |
| More than 10 years | 70 | 23% |
| **Marital Status** | | |
| Single | 95 | 31.3% |
| Currently Married | 207 | 68.1% |
| Separated/Divorced | 1 | 0.3% |
| Prefer Not To Say | 1 | 0.3% |
| **Psychiatric History** | | |
| Present | 29 | 9.6% |
| Not Present | 275 | 90.4% |
| **Depression** | | |
| Present | 76 | 25% |
| Absent | 228 | 75% |
| **Anxiety** | | |
| Present | 97 | 31.9% |
| Absent | 207 | 68.1% |
| **PTSD** | | |
| Present | 48 | 15.8% |
| Absent | 256 | 84.2% |

psychiatric illness among respondents. The study participants who were females, from the surgery and allied departments, frontline workers, junior staff members, and those having a history of psychiatric illness were observed to have higher statistically significant Median (IQR) scores (Table 2).

## Severity of symptoms against background characteristics

Comparatively, a higher proportion of women than men and junior staff than the senior staff reported more severe symptoms of depression and anxiety. Likewise, a higher proportion of respondents who had a history of psychiatric illness

Table 2. Depression, anxiety, and PTSD scores across background categories.

| Scale | Total Score Median (IQR) | Gender | | | Department/Specialty | | | Line of Work During the Pandemic | | | Staff's Position/Seniority | | | Psychiatric History | | |
|---|---|---|---|---|---|---|---|---|---|---|---|---|---|---|---|---|
| | | Median (IQR) | | p value | Median (IQR) | | p value | Median (IQR) | | p value | Median (IQR) | | p value | Median (IQR) | | p value |
| | | Men | Women | | Medicine & Allied | Surgery & Allied | | Frontline | Second line | | Junior Staff | Senior Staff | | Present | Not present | |
| **PHQ-9 Depression symptoms score** | 5 (2-9.5) | 4 (1-9) | 6 (2-11) | 0.004* | 4 (1-9) | 8 (4-12) | 0.001* | 5.5 (2-10) | 3.5 (0-9) | 0.035* | 6 (2-11) | 2 (0-6) | <0.001* | 9 (6-14) | 5 (1-9) | <0.001* |
| **GAD-7, anxiety symptoms score** | 4 (0-7) | 3 (0-6) | 5 (1-9) | 0.001* | 3 (0-7) | 4 (2-9) | 0.036* | 4 (1-8) | 2 (0-6.5) | 0.009* | 5 (2-9) | 1 (0-4) | <0.001* | 7 (3-10) | 3 (0-7) | 0.004* |
| **IES-R, Total PTSD symptoms score** | 10 (1-23) | 8.5 (1-22) | 13 (1-29) | 0.041* | 10 (1-23) | 16 (5-31) | 0.014* | 12 (2-24) | 5 (0-21) | 0.030* | 13 (2-25) | 5 (0-18) | <0.001* | 24 (10-42) | 9 (1-22) | <0.001* |
| *Intrusion Subscale score* | 4 (0-8) | 2.5 (0-8) | 5 (1-10) | 0.030* | 3 (0-8) | 5.5 (1-9) | 0.050* | 4.5 (0.5-9) | 2 (0-6) | 0.014* | 5 (1-9.5) | 2 (0-6) | 0.002* | 9 (2-15) | 3 (0-8) | <0.001* |
| *Avoidance Subscale score* | 4 (0-10) | 3 (0-8) | 4 (0-11) | 0.035* | 3 (0-8) | 6 (3-14) | 0.002* | 4 (0-10) | 3 (0-10) | 0.191 | 4 (0.5-11) | 2 (0-8) | <0.001* | 8 (4-12) | 3 (0-8) | <0.001* |
| *Hyper-arousal Subscale score* | 2 (0-6) | 2 (0-6) | 3 (0-7) | 0.071 | 2 (0-6) | 3 (1-6) | 0.016* | 3 (0-7) | 1 (0-6) | 0.010* | 3 (0-7) | 1 (0-4) | <0.001* | 6 (3-12) | 2 (0-6) | <0.001* |

*P: statistically significant. †PHQ-9: Patient Health Questionnaire-9. ‡GAD-7: Generalized Anxiety Disorder-7. §IES-R: Impact of Events Scale-Revised. ¶PTSD: Post-traumatic Stress Disorder. #IQR: Inter Quartile Range.

**Table 3. Severity categories of depression, anxiety, and PTSD and associated factors.**

| Severity Category | Overall Severity f (%) | Gender | | | Department/Specialty | | | Line of Work during the Pandemic | | | Staff's Position/Seniority | | | Psychiatric History | | |
|---|---|---|---|---|---|---|---|---|---|---|---|---|---|---|---|---|
| | | Men f (%) | Women | p value | Medicine & Allied f(%) | Surgery & Allied | P value | Front line f (%) | Second line | p value | Senior staff f (%) | Junior Staff | p value | Present f (%) | Not Present | p value |
| **PHQ-9, Depression Symptoms Severity** | | | | | | | | | | | | | | | | |
| Normal | 140 (46.1) | 88 (53) | 52 (37.7) | 0.041* | 119 (51.1) | 20 (28.6) | 0.001* | 96 (42.1) | 44 (57.9) | 0.041* | 54 (63.5) | 86 (39.3) | 0.001* | 4 (13.8) | 136 (49.6) | 0.003* |
| Mild | 88 (28.9) | 44 (26.6) | 44 (31.9) | | 63 (27) | 25 (35.7) | | 73 (32) | 15 (19.7) | | 19 (22.4) | 69 (31.5) | | 13 (14.8) | 75 (27.6) | |
| Moderate | 34 (11.2) | 17 (10.2) | 17 (12.3) | | 24 (10.3) | 10 (19.3) | | 29 (12.7) | 5 (6.6) | | 7 (8.2) | 27 (12.3) | | 5 (17.2) | 28 (10.2) | |
| Severe | 42 (13.8) | 17 (10.2) | 25 (18.1) | | 27 (11.6) | 15 (21.4) | | 30 (13.2) | 12 (15.8) | | 5 (5.9) | 37 (16.9) | | 7 (24.1) | 35 (12.8) | |
| **GAD-7, Anxiety Symptoms Severity** | | | | | | | | | | | | | | | | |
| Normal | 172 (56.6) | 108 (65.1) | 64 (46.4) | 0.001* | 135 (57.9) | 36 (51.4) | 0.892 | 119 (52.2) | 53 (69.7) | 0.061 | 64 (75.3) | 108 (49.3) | 0.001* | 12 (41.4) | 160 (58.4) | 0.136 |
| Mild | 75 (24.4) | 35 (21.1) | 40 (29) | | 57 (24.5) | 18 (25.7) | | 63 (27.6) | 12 (15.8) | | 14 (16.5) | 61 (27.9) | | 7 (24.1) | 67 (24.5) | |
| Moderate | 35 (11.5) | 18 (10.8) | 17 (12.3) | | 26 (11.2) | 9 (12.9) | | 28 (12.3) | 7 (9.2) | | 5 (5.9) | 30 (13.7) | | 6 (20.7) | 29 (10.6) | |
| Severe | 22 (7.2) | 5 (3) | 17 (12.3) | | 15 (6.4) | 7 (10) | | 18 (7.9) | 4 (5.3) | | 2 (2.4) | 20 (9.1) | | 4 (13.8) | 18 (6.6) | |
| **IES-R, PTSD Symptoms Severity** | | | | | | | | | | | | | | | | |
| No PTSD | 230 (75.7) | 133 (80.1) | 97 (70.3) | 0.078 | 117 (76) | 52 (74.3) | 0.063 | 168 (73.7) | 62 (81.6) | 0.502 | 75 (88.2) | 155 (70.8) | 0.013* | 12 (41.4) | 218 (79.6) | <0.001* |
| Partial PTSD | 26 (8.6) | 15 (9) | 11 (8) | | 25 (10.6) | 1 (1.4) | | 21 (9.2) | 5 (6.6) | | 4 (4.7) | 22 (10) | | 7 (24.1) | 18 (6.6) | |
| Probable PTSD | 7 (2.3) | 3 (1.8) | 4 (2.9) | | 6 (2.6) | 1 (1.4) | | 5 (2.2) | 2 (2.6) | | 0 | 7 (3.2) | | 1 (3.4) | 6 (2.2) | |
| PTSD Present | 41 (13.5) | 15 (9) | 26 (18.8) | | 25 (10.7) | 16 (22.9) | | 34 (14.9) | 7 (9.2) | | 6 (7.1) | 35 (16) | | 9 (31.0) | 32 (11.7) | |

*P: statistically significant. †PHQ-9: Patient Health Questionnaire-9. ‡GAD-7: Generalized Anxiety Disorder-7 §IES-R Impact of Events Scale-Revised. ¶PTSD: Post-Traumatic Stress Disorder.

reported moderate (17.2% vs. 10.2%) and severe symptoms of depression (24.1% vs. 12.8%) and PTSD (37% vs. 11.1%) than those who had no significant psychiatric history (Table 3).

We did not find a significant statistical difference in anxiety severity across departments, frontline or second-line working positions, and history of psychiatric illness. The severity of PTSD had no statistically significant differences among men or women, staff on the frontline or second line against COVID-19, and staff from the Surgery & allied or Medicine & allied departments (Table 3).

## Discussion

In our study, 15.8% of the doctors suffered from PTSD, 25% from depression, and 31.9% from anxiety symptoms, even during the fourth COVID-19 wave, indicating a considerable burden of psychological distress in healthcare providers, even more than a year after the confirmation of the first case of COVID-19 pneumonia in Pakistan in February 2020 [20]. Infectious disease outbreaks and epidemics have a substantial negative impact on the psychosocial well-being of healthcare professionals. Research showed that a high proportion of healthcare providers suffered from high levels of emotional distress during the 2003 SARS epidemic, such as approximately 29–35% of healthcare workers were found to suffer from traumatic stress in Toronto, Canada [21]. Anxiety, depression, post-traumatic stress disorder, and burnout in healthcare providers have been reported in various viral outbreaks, viz. MERS, Ebola virus, and H1N1 influenza [22,23].

Healthcare workers faced significantly higher physical and psychological risks during the COVID-19 pandemic in comparison to the general population. They were at higher risk of infection due to relatively direct exposure and insufficient protective gear. Contributing factors to psychological distress included excessive work, perceived stigmatization, necessary quarantine, and, in some cases, inadequate psycho-social support [24]. A multicenter study from Pakistan found that 28.8% of nurses, doctors, and allied health staff had symptoms of PTSD, 22.75% had depression, and 33.75% had clinically significant anxiety during the early wave of the COVID-19 pandemic [25]. A study from a teaching hospital in Pakistan reported 27.9%, 30%, and 36.2% prevalence of acute stress disorder, depression, and anxiety, respectively, in early 2020, indicating the rates of depression and anxiety were comparable to our findings, but a relatively higher prevalence of traumatic stress during the first wave [26].

Our study revealed higher scores for anxiety as compared to depression, which is similar to previous reports [27,28]. This level of anxiety among doctors, as in our sample, could be attributed to factors like emergency deployments of untrained doctors to COVID-19 wards, uncertainties related to working hours, longer shifts, lack of proper resting places, limited protective gear, and fear of infection. The comparatively lower prevalence of PTSD and depression in our study, contrary to that reported from Pakistan in the initial peaks, might be linked to the COVID-19 vaccination drive in Pakistan, which started in early 2021 and covered front-line healthcare workers in the initial phase [29]. This may also be associated with less exposure of healthcare providers to critical patients, severe illness, and mortality; better personal protection; psychological adaptation, and individual and system-related resilience factors during the later stages. COVID-19 vaccines have been tremendously effective in reducing the incidence of COVID-19 pneumonia, hospitalization, and mortality worldwide [30]. Research across the United States and Europe has reported an association between COVID-19 vaccination and a decline in psychological distress among healthcare providers, including doctors [31,32].

Moreover, differences in comparative levels of PTSD among doctors across the different phases of the COVID-19 pandemic may be attributed to methodological variations in various studies, particularly in sampling and data collection procedures. Research has also highlighted the cultural and ethno-racial differences in the psychological impact of the COVID-19 pandemic across the world [33,34]. Therefore, it is crucial to understand the role of sociocultural differences (stigma, coping strategies, and health-seeking behaviors) among doctors from different cultural backgrounds while comparing differences in the mental health impact.

During the pandemic, various demographic, occupational, and psychosocial factors were found to be contributing towards the psychological burden in healthcare providers. These included age, gender, exposure to infected individuals

and periods of quarantine, job roles, work experience, and psycho-social factors such as coping styles and community stigmatization [35]. A significant proportion of health workers involved in the pandemic response, particularly those on the front-lines, young women, and those previously diagnosed with mental health conditions such as depression, anxiety, and post-traumatic stress disorder, were at an increased risk [36]. The body of available literature also highlights the fear and risk of getting infected and passing on the infection, a lack of protective gear, and inadequate social support as potential risk factors for mental distress in healthcare personnel combating the pandemic [37,38]. These differences in healthcare infrastructure, availability of healthcare resources, and perceived organizational and administrative support could be the potential contributing factors to differences in psychological distress in doctors across the high-income and lower-middle, and low-income countries. Various factors that protected the healthcare providers from the adverse psychological impact of the pandemic included having adequate healthcare resources, taking protective measures, and being familiar with the latest evidence and clinical practice guidelines [39].

In our study, female and junior-level doctors, including interns, residents, and medical officers, those working on the front-lines and had a psychiatric history, had comparatively higher, statistically significant scores of anxiety, depression, and PTSD. Alonso J, et al reported from Spain that the odds of suffering from a mental illness during the pandemic were found to be higher in female healthcare workers, those having prior psychiatric history, or were caring for COVID-19 patients [39]. Lasalvia A, and others also reported female gender, direct care of COVID-19 patients, and pre-existing mental health issues as the risk factors of psychopathology in healthcare workers of North-East Italy during the pandemic [40].

We found that providing direct care to COVID-19 patients was an independent risk factor for depression but not anxiety and PTSD. These results contradicted a multicenter study from China in which frontline workers were 1.40 times more prone to suffer from Post-Traumatic Stress Disorder (PTSD) and depression [41]. Interestingly, our study also found that neither gender nor department predicted or influenced the severity of PTSD. However, a junior position and a prior history of mental illness were independently linked to higher PTSD severity. Our results contrast with a Norwegian study, which identified direct or indirect care of COVID-19 patients as a risk factor for severe PTSD, in addition to pre-existing depressive and anxiety disorders [42]. Askanew, S et al from Ethiopia, also reported a significant association of PTSD with a history of psychiatric illness. They reported a prevalence of PTSD as high as 55.1%, indicating a high proportion of healthcare workers battling traumatic stress in low-income countries [43].

Junior healthcare providers, particularly doctors, tend to face more workload and extended working hours, hence are more exposed to intense and stressful work environments, which predispose them to psychological distress. Interestingly, we found that being female was linked to a higher likelihood of depression and anxiety, though this association did not extend to PTSD. We also found that junior doctors were at a higher risk of anxiety, depression, and PTSD. Corresponding to our findings that females and junior staff were at higher odds of suffering from depressive and anxiety disorders, a study from Spain also reported that women and the personnel doing 12–24-hour shift work were more prone to anxiety and depressive symptoms, whilst resilience and a sense of self-satisfaction acted as protective factors [44].

Much of the research on the psychological impact of the pandemic on healthcare staff was conducted during the acute and crisis phase of it. In contrast, our study was conducted during the fourth wave of the pandemic; therefore, it provided valuable insights into the enduring and longer-term psychological impact and evolving coping mechanisms of doctors. Owing to the cross-sectional nature of our study, it was not possible to establish the temporal associations and cause-and-effect relationships between the risk factors and mental health outcomes. Moreover, the longitudinal and delayed trajectories of mental health outcomes of the pandemic were also not elicited due to the cross-sectional design. As the focus of our study was limited to doctors, the mental health experiences of other professionals, including nursing and paramedical staff, could not be captured. The findings of our study on a small sample from one city may not reflect the experiences of the broader healthcare population, hence limiting the generalizability of our findings. Furthermore, there was a potential risk of sampling bias and the probability of over- or under-representation of doctors with adverse mental health impacts of the pandemic in the context of our non-probability convenience sample. Nonetheless, we acknowledge that a multistage

sampling strategy with a random sampling technique for hospital selection in the first stage and quota sampling for gender, front-line work, and seniority levels in the second stage would have improved representativeness and generalizability and reduced the selection and sampling bias. However, the logistical and resource constraints made probability sampling particularly stratified and systematic random sampling techniques unfeasible for our study.

We would also acknowledge the potential language bias in our study due to the use of English versions of the PHQ-9, GAD-7, and IES-R in our study. Although doctors in Pakistan are generally fluent in English and the English versions of the scales are readily used in Pakistan, using validated Urdu translations could have enhanced cultural sensitivity. We further acknowledge a statistical limitation in not applying regression analysis and the likely retention of confounding factors. While logistic regression does not require normally distributed data, applying binary or ordinal logistic models in our context of non-normal distribution would require collapsing or restructuring the severity categories of three outcomes of depression, anxiety, and PTSD, leading to a loss of important information. Therefore, we relied on non-parametric methods relevant to the data distribution.

Our study complements the existing literature and highlights the crucial role of early detection and effective management of psychological distress among doctors in times of public health crisis. It also emphasizes that the mental well-being of healthcare personnel should be considered an essential component of disaster preparedness and management. Effective individual and organizational-level interventions are needed to prevent, identify, and treat psychological distress and mental health problems in healthcare providers during high-stress situations. Adequate personal protection, better communication, critical care training, flexible and humane work schedules, and ongoing mental health monitoring and support even after the end of a crisis are a few compelling strategies to prevent and effectively manage mental health crises in healthcare providers during such public health disasters. The critical issue of the mental well-being of healthcare providers continues to need attention even after the end of the pandemic. Other key strategies to improve it include resilience-building training, a supportive work environment, implementation of stress management programs, and provision of psychological support through confidential counseling, therapy, and support group services [45]. Finally, it is crucial to advocate for systemic policy changes that support the mental health of healthcare providers and address issues such as workload, staffing, work-hour regulations, and administrative burdens, aimed at the sustainability of public healthcare infrastructure and workforce retention. These strategies have also been recommended in prior research for reducing the risk of occupational moral injury and its unfavorable mental health consequences for healthcare workers during the COVID-19 pandemic [46].

## Conclusion

This study showed that a substantial number of doctors in our sample continued to experience the symptoms of depression, anxiety, and PTSD even during the fourth wave of the COVID-19 pandemic. This underscores the enduring psychological impact of prolonged healthcare crises within our study context and may provide valuable insights into the urgent need to prioritize the mental well-being of the essential workforce, including doctors, in both the acute phases of public health crises and the long-term aftermath. However, the cross-sectional design limits inferences regarding causality or the enduring and long-term psychological impact of the COVID-19 pandemic. In addition, the extrapolation of findings to a policy context should be interpreted as speculative and not prescriptive, in light of the study's limited generalizability and the absence of multivariable adjustment. While broad policy implications cannot be directly inferred, the findings may still offer preliminary insights relevant to discussions on disaster preparedness and response, particularly with respect to integrating psychological well-being alongside physical health considerations in a disaster response. We recommend that future research focusses on identifying practicable interventions tailored to the needs of healthcare providers, including doctors, for the prevention and management of mental health challenges within this essential workforce, particularly during times of disaster. Further longitudinal research, using larger and more representative samples and multivariable analyses, is needed to better understand the determinants of mental health outcomes and the effective interventions required to

support doctors and other healthcare providers during times of disaster. This may include a focus on hospital-level institutional and administrative reforms through a conducive workplace environment, access to mental health supports for staff, and policy-level structural reformation and effective overhaul of systemic issues in the Public healthcare infrastructure in Pakistan.

## Author contributions

**Conceptualization:** Madah Fatima, Nazish Imran, Irum Aamer, Bilquis Shabbir.

**Data curation:** Madah Fatima, Nazish Imran, Irum Aamer, Somia Iqtadar, Bilquis Shabbir.

**Formal analysis:** Madah Fatima, Nazish Imran.

**Investigation:** Madah Fatima, Nazish Imran.

**Methodology:** Madah Fatima, Nazish Imran, Irum Aamer, Somia Iqtadar, Bilquis Shabbir.

**Resources:** Madah Fatima, Nazish Imran, Irum Aamer, Somia Iqtadar, Bilquis Shabbir.

**Supervision:** Nazish Imran.

**Writing – original draft:** Madah Fatima.

**Writing – review & editing:** Madah Fatima, Nazish Imran, Irum Aamer, Somia Iqtadar, Bilquis Shabbir.

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
