## [Decision Letter · Decision Letter 0]

14 May 2025

PMEN-D-25-00130

Disaster and Distress: The Double Burden.

Depression, Anxiety, and Post-Traumatic Stress Disorder in Healthcare Providers during the COVID-19 Pandemic in Pakistan

PLOS Mental Health

Dear Dr. Fatima,

Thank you for submitting your manuscript to PLOS Mental Health. After careful consideration, we feel that it has merit but does not fully meet PLOS Mental Health’s publication criteria as it currently stands. Therefore, we invite you to submit a revised version of the manuscript that addresses the points raised during the review process.

We look forward to receiving your revised manuscript.

Kind regards,

Annesha Sil, Ph.D.

Staff Editor

PLOS

Journal Requirements:

1. We noticed you have some minor occurrence of overlapping text with the following previous publication(s), which needs to be addressed:

- doi: 10.3389/fpsyt.2023.1244055

- DOI: 10.21203/rs.3.rs-2029236/v1

- doi: 10.1016/S2215-0366(20)30168-1

In your revision ensure you cite all your sources (including your own works), and quote or rephrase any duplicated text outside the methods section. Further consideration is dependent on these concerns being addressed.

Additional Editor Comments (if provided):

Reviewers' comments:

Reviewer's Responses to Questions

**Comments to the Author**

1. Does this manuscript meet PLOS Mental Health’s publication criteria?

Reviewer #1: Yes

Reviewer #2: Yes

2. Has the statistical analysis been performed appropriately and rigorously?

Reviewer #1: I don't know

Reviewer #2: Yes

3. Have the authors made all data underlying the findings in their manuscript fully available (please refer to the Data Availability Statement at the start of the manuscript PDF file)?

Reviewer #1: Yes

Reviewer #2: Yes

4. Is the manuscript presented in an intelligible fashion and written in standard English?

Reviewer #1: Yes

Reviewer #2: Yes

Reviewer #1: • The title of the study states “Depression, Anxiety, and Post-Traumatic Stress Disorder in Healthcare Providers”. However, it only examines the psychological consequences of the pandemic on doctors, and no other healthcare providers such as nurses etc. Therefore, it may be more appropriate to change the title to reflect that the study was done on doctors only.

• This study was conducted across tertiary care hospitals in Lahore. How many tertiary care hospitals were included in the study? Did they include all tertiary care hospitals in Lahore?

• Please describe the sample size calculation

• How were the participants recruited?

• Please give references for the validation of PHQ-9, GAD-7 and IES-R in Pakistani population.

• The authors state that the IES-R measures PTSD on a 4-point Likert scale. It should be corrected as a 5-point Likert scale (0-4).

• The authors state that a score of 39 or above indicates confirmed PTSD. However, IES-R is not a diagnostic instrument, it is commonly employed as a screening measure to assess the presence and severity of PTSD symptoms. I could not find literature interpreting a score of >39 as “confirmed PTSD””.

• Was the Urdu version or the English version of the questionnaires used?

• The authors state that “Comparatively, a higher proportion of women than men and junior staff than the senior staff reported more severe symptoms of depression and anxiety” (line 229-230). Then they go on to say that “there were no statistically significant differences in the severity of anxiety symptoms across gender” (line 234). These two statements about anxiety are contradictory.

• There are numerous studies done on mental health of healthcare workers in Pakistan during COVId-19. The discussion should describe what this article adds to the literature, that has not already been published.

• There are numerous recent articles that are relevant to this study that have not been referenced or cited.

Reviewer #2: Thank you for the opportunity to review your manuscript on the critical issue of mental health among healthcare providers during the COVID-19 pandemic in Pakistan. Your study addresses a significant topic, and the employment of validated tools such as the PHQ-9, GAD-7, and IES-R is commendable. However, several areas need improvement to enhance the manuscript's clarity and robustness.

The use of non-probability convenient sampling is a notable limitation that affects the generalizability of your findings. While you have acknowledged this limitation, a more comprehensive discussion on how this sampling method might have influenced your results would add depth to your manuscript. Additionally, although alternative sampling strategies were not feasible in your study, a brief consideration of how these methods could have potentially provided a more representative sample would be valuable.

In terms of statistical analysis, the rationale behind choosing specific non-parametric tests for different comparisons needs further elaboration. Providing a clear justification for the use of the Mann-Whitney U test, Kruskal-Wallis test, and Chi-square test in your analyses would clarify their appropriateness. Furthermore, it is crucial to address whether any adjustments for multiple comparisons were made, given the number of statistical tests conducted. Discussing methods like the Bonferroni correction could help mitigate the risk of Type I errors.

Your discussion section effectively compares results with existing literature, but a more critical evaluation of discrepancies, particularly regarding the prevalence of PTSD and influencing factors, would be beneficial. Beyond attributing differences solely to the vaccination drive, consider exploring cultural, systemic, or methodological factors that might contribute to these variations. This approach would offer a more nuanced interpretation of your findings in the context of global research.

There are several grammatical errors and awkward phrasings that impact readability. For instance, the sentence "The economic downturns, shutdown of businesses, and unemployment attributed to the disease burden, disaster mitigation and disease containment measures such as lockdowns impacted the whole world," could be revised to "The economic downturns, shutdown of businesses, and unemployment resulting from the disease burden and containment measures such as lockdowns impacted the whole world." A thorough grammatical review would enhance the manuscript's clarity and professionalism.

Finally, integrating the tables more effectively into the narrative would improve the presentation of your results. Highlighting key findings from the tables within the text would allow readers to quickly grasp the main outcomes and their implications. Ensuring consistency in reporting percentages and decimal places throughout the manuscript will also contribute to a more polished presentation of the data.

Overall, your study provides meaningful insights into an important topic, but revising these aspects will improve methodological transparency, analytical rigor, and manuscript clarity. I recommend addressing these issues comprehensively for a stronger manuscript.

**Do you want your identity to be public for this peer review?** For information about this choice, including consent withdrawal, please see our Privacy Policy

Reviewer #1: No

Reviewer #2: No

---

## [Decision Letter · Decision Letter 1]

20 Oct 2025

PMEN-D-25-00130R1

Disaster and Distress: The Double Burden.Depression, Anxiety, and Post-Traumatic Stress Disorder in Doctors during the COVID-19 Pandemic in Pakistan

PLOS Mental Health

Dear Dr. Fatima,

Thank you for submitting your manuscript to PLOS Mental Health. After careful consideration, we feel that it has merit but does not fully meet PLOS Mental Health’s publication criteria as it currently stands. Therefore, we invite you to submit a revised version of the manuscript that addresses the points raised during the review process.

*The manuscript addresses a relevant and timely topic, exploring mental health among physicians during the COVID-19 pandemic in a lower-middle-income country. Although it employs standardized instruments (PHQ-9, GAD-7, IES-R) previously validated in Pakistan, the limitations section fails to discuss potential bias arising from the use of English versions of these tools, despite the availability of validated Urdu versions.*

*The conclusion would be stronger and more persuasive if multivariable models were applied to control for potential confounding factors. This analytical limitation should be explicitly acknowledged and addressed in the revision.*

We look forward to receiving your revised manuscript.

Kind regards,

João Silvestre Silva-Junior

Academic Editor

PLOS Mental Health

---

## [Decision Letter · Decision Letter 2]

7 Jan 2026

PMEN-D-25-00130R2

Disaster and Distress: The Double Burden. Depression, Anxiety, and Post-Traumatic Stress Disorder in Doctors during the COVID-19 Pandemic in Pakistan

PLOS Mental Health

Dear Dr. Fatima,

Thank you for submitting your manuscript to PLOS Mental Health. After careful consideration, we feel that it has merit but does not fully meet PLOS Mental Health’s publication criteria as it currently stands. Therefore, we invite you to submit a revised version of the manuscript that addresses the points raised during the review process.

Regarding the response to the methodological concerns, their justification is contextually reasonable, and the transparent discussion of this limitation in the manuscript is acceptable. Concerning the absence of multivariable analyses, it is a methodological standpoint and the limitation is now explicitly framed in the manuscript.

With respect to the conclusion, it is broadly consistent with the study findings; however, some statements would benefit from being slightly tempered to better reflect the cross-sectional design, limited generalizability, and lack of multivariable adjustment. In particular, references to long-term or enduring impact and broad policy implications should be clearly framed as potential or speculative rather than directly inferred from the data.

We look forward to receiving your revised manuscript.

Kind regards,

João Silvestre Silva-Junior, MD MSc PhD

Academic Editor

PLOS Mental Health

---

## [Editor Report · Decision Letter 3]

14 Jan 2026

Disaster and Distress: The Double Burden. Depression, Anxiety, and Post-Traumatic Stress Disorder in Doctors during the COVID-19 Pandemic in Pakistan

PMEN-D-25-00130R3

Dear Dr. Fatima,

We are pleased to inform you that your manuscript 'Disaster and Distress: The Double Burden. Depression, Anxiety, and Post-Traumatic Stress Disorder in Doctors during the COVID-19 Pandemic in Pakistan' has been provisionally accepted for publication in PLOS Mental Health.

Best regards,

João Silvestre Silva-Junior, MD MSc PhD

Academic Editor

PLOS Mental Health